# The Role of the Iron Protoporphyrins Heme and Hematin in the Antimalarial Activity of Endoperoxide Drugs

**DOI:** 10.3390/ph15010060

**Published:** 2022-01-04

**Authors:** Helenita C. Quadros, Mariana C. B. Silva, Diogo R. M. Moreira

**Affiliations:** Fundação Oswaldo Cruz (Fiocruz), Instituto Gonçalo Moniz, Salvador 40296-710, Bahia, Brazil; helenita_quadros@hotmail.com (H.C.Q.); marii.borges@hotmail.com (M.C.B.S.)

**Keywords:** *Plasmodium*, endoperoxide, artemisinin, hematin, heme, hemozoin, adducts, heme speciation

## Abstract

*Plasmodium* has evolved to regulate the levels and oxidative states of iron protoporphyrin IX (Fe-PPIX). Antimalarial endoperoxides such as 1,2,4-trioxane artemisinin and 1,2,4-trioxolane arterolane undergo a bioreductive activation step mediated by heme (FeII-PPIX) but not by hematin (FeIII-PPIX), leading to the generation of a radical species. This can alkylate proteins vital for parasite survival and alkylate heme into hematin–drug adducts. Heme alkylation is abundant and accompanied by interconversion from the ferrous to the ferric state, which may induce an imbalance in the iron redox homeostasis. In addition to this, hematin–artemisinin adducts antagonize the spontaneous biomineralization of hematin into hemozoin crystals, differing strikingly from artemisinins, which do not directly suppress hematin biomineralization. These hematin–drug adducts, despite being devoid of the peroxide bond required for radical-induced alkylation, are powerful antiplasmodial agents. This review addresses our current understanding of Fe-PPIX as a bioreductive activator and molecular target. A compelling pharmacological model is that by alkylating heme, endoperoxide drugs can cause an imbalance in the iron homeostasis and that the hematin–drug adducts formed have strong cytocidal effects by possibly reproducing some of the toxifying effects of free Fe-PPIX. The antiplasmodial phenotype and the mode of action of hematin–drug adducts open new possibilities for reconciliating the mechanism of endoperoxide drugs and for malaria intervention.

## 1. Affinity between Peroxides and Iron Protoporphyrins

The importance of artemisinin, an endoperoxide sesquiterpene lactone, in antimalarial therapy has shaped the way medicinal chemists used to look at heme (ferrous protoporphyrin IX, Fe^II^-PPIX). From a simple reactant and cofactor often used in biological assays [1], heme is now recognized as a drug activator, transporter, and target for antimalarial endoperoxides. This began after a seminal paper in 1991 showed that artemisinin undergoes molecular activation in *Plasmodium* parasite cells, which at that time led to the isolation of an artemisinin conjugate bound to a cofactor presumed to be an iron protoporphyrin [2]. Chemical synthesis has shown that heme initiates reductive activation of the artemisinin’s peroxide group via the iron Fenton reaction, resulting in oxy-radical formation, which further rearranges into a *C*-centered radical, ultimately alkylating heme through one of the meso positions of the porphyrin macrocycle by an intramolecular reaction that produces iron protoporphyrin-artemisinin adducts [3]. The iron atom present in the iron protoporphyrin-artemisinin adducts formed under biogenic conditions (i.e., inside parasites) and synthetic conditions is likely to be in the ferric state. These adducts are hereinafter referred to as hematin–drug adducts [4,5]. In addition to the alkylation of heme, the bioreductive activation of artemisinin by cellular heme can result in radical-induced protein alkylation, an event resulting in the inactivation of a variety of biological processes necessary for parasite survival and growth [6,7] (Figure 1).

The chemistry of the bioreductive activation of peroxides by heme has been known for decades [8]. It is also known that this activation is mediated by ferrous heme but not by ferric hematin [9]. Despite this, the precise extent and relevance of this activation for the antimalarial activity of peroxides continue to be matters of debate. Today, there is compelling evidence that the bioreductive activation of peroxides by heme is not solely a drug activation process. The heme-mediated activation of antimalarials has deep ties with the mechanism of antimalarial activity, the spectrum of activity for multiple stages of the *Plasmodium* life cycle, and the mechanism of parasite resistance to antimalarials. Comprehensive reviews of endoperoxides for malaria therapy [10], their structure–activity relationships [11,12], and the chemical mechanism of the redox chemistry involving heme [3,13] have been addressed elsewhere. The aim of this paper is to address the most compelling roles of Fe-PPIXs as molecular targets and bioreductive activators of antimalarial peroxide drugs.

## 2. Introduction to Heme Detoxification

Other pathogens have evolved to assimilate heme into specific metabolic pathways [14,15] or to appeal heme oxygenases to enzymatically convert heme into harmless but recyclable species [16]. This contrasts with *Plasmodium*, which relies on hematin biomineralization for the heme detoxification process, converting soluble and toxic Fe-PPIX species into harmless but unrecyclable hemozoin (Hz) crystals.

Heme detoxification begins with the digestion of hemoglobin inside the digestive vacuole (DV) during the intraerythrocytic blood stage of *Plasmodium* (referred to as the asexual blood stage); this peaks at the trophozoite stage, when intense digestion of hemoglobin takes place. Efficient hemoglobin digestion is enabled by a combination of aspartic proteinases (plasmepsins) and cysteine proteinases (falcipains) in an acidic environment [17,18,19]. The hydrophobic pocket of hemoglobin is important for stabilizing the ferrous iron presented in heme. As such, the digestion of hemoglobin and subsequent release of heme enable its rapid oxidization with molecular oxygen to its counterpart ferric form (ferric protoporphyrin IX, Fe^III^-PPIX, hematin) [20]. Then, the most astonishing process occurs, which is the dimerization of hematin through the reciprocal coordination of iron and propionate moieties into β-hematin and its subsequent crystallization into Hz [21,22]. Coincidentally, the DV and its content (Hz) are masterpieces of heme detoxification and hallmarks of *Plasmodium* biology [23,24] (Figure 2).

*Plasmodium* heme detoxification is only possible because the DV, where it takes place, is an organelle specialized in assembling small vesicles filled with undigested hemoglobin from the erythrocyte’s cytoplasm (cytostomes). After the cytostome assemble in the DV and hemoglobin and lipids are internalized, an aqueous and lipidic environment is formed where Hz crystals grow [25,26]. Hz’s unique brown and birefringent crystallinity can be detected and monitored by imaging techniques, and owing to its relatively simple composition, Hz can be synthetically reproduced in the laboratory. In fact, Hz crystals can be studied in situ by a variety of biomedical methods, from simple optical microscopy using Giemsa stain to polarized light microscopy (for which no staining is required), and even flow cytometry [27,28,29]. Nowadays, there are high-resolution imaging techniques for studying Hz, such as atomic force microscopy and transmission X-ray microscopy; these have been decisive in elucidating each step of Hz nucleation, growth, and orientation [30].

Since the first observations that treating rodents and nonhuman primates infected by *Plasmodium* with 4-aminoquinoline chloroquine affects the morphology of DV and reduces Hz formation [31], scientists have studied the mechanism of drug blockade targeting the heme detoxification [32]. Heme detoxification is sensitive to drug interventions, but importantly, it remains practically unaltered even when drug-resistant parasites emerge, indicating that it is a powerful target for antimalarial drugs [33,34]. In terms of suppression by drug intervention, it can be categorized into two main mechanisms: drugs targeting the soluble Fe-PPIX (sometimes referred to as the non-crystalline form) through iron coordination chemistry to suppress the formation of crystalline components [24] and drugs that bind by adsorption mechanisms to the Hz crystal surfaces and suppress crystal elongation [35]. Both mechanisms, schematically depicted in Figure 2, can produce an increase in heme levels that are toxic for parasite cells (heme toxification); regardless of which mechanism is employed, the Fe-PPIX-drug interaction is certainly a key event for suppressing heme detoxification.

Not surprisingly, considering the importance of Hz as a hallmark of *Plasmodium* biology and drug target, the scientific community has devoted efforts to studying whether antimalarial peroxides can suppress *Plasmodium* heme detoxification. Prior to this, a decisive method for addressing this question was the development of β-hematin inhibitory activity, an assay using synthetic hematin hydroxide (OH-Fe^III^-PPIX) obtained from hemin chloride (Cl-Fe^III^-PPIX), which, when exposed to a buffered acid yields β-hematin crystals that present spontaneous crystal growth with a chemical signature identical to that of Hz crystals found in parasite cells [36,37]. Unlike the antimalarial 4-aminoquinolines, such as chloroquine and amodiaquine, which strongly suppress β-hematin crystal formation, representative antimalarial endoperoxides such as artemisinin and dihydroartemisinin do not [38]. These results indicate that artemisinins in general do not target soluble Fe^III^-PPIX to suppress the formation of β-hematin crystals. A second mechanism, where a drug binds by adsorption mechanisms to the surface of growing β-hematin crystals and suppresses crystal elongation, also indicated that artemisinin does not interfere with β-hematin crystal elongation [35,39]. All these findings, validated by independent research groups [40,41], converged to the notion that antimalarial peroxides do not interfere with hematin mineralization.

Subsequently, it became clear that the absence of β-hematin inhibitory activity for antimalarial peroxides is due to a lack of binding affinity of these drugs for hematin [9]. Considering that hematin, but not heme, is the only iron species responsible for hematin biomineralization, it became apparent for a while that peroxides do not affect *Plasmodium* heme detoxification, at least not by a canonical mechanism where a drug, such as chloroquine, binds to soluble hematin and suppresses the onset of the crystal formation, adsorbing onto the surface of growing crystals and thereby preventing their elongation [35,42].

Apart from cell-free models with soluble hematin and β-hematin crystals, studies of *Plasmodium* heme detoxification in cell cultures have been employed in short-pulse treatments (<8 h of drug exposure) for the asexual blood stage of *Plasmodium falciparum* (*P. falciparum*). Hung et al. [43] made use of a radiolabeled artemisinin and protocols for extracting Hz to show that artemisinin treatment was associated with the Hz content, likely in the form of hematin–artemisinin adducts, denoting a possible fate of artemisinin to interact with Hz. However, using mass spectroscopy, two independent research groups were unable to identify endoperoxide and its hematin adducts adsorbed on Hz from parasites treated with artemisinin, artesunate or dihydroartemisinin in a concentration range from 500 to 5000 nM [41,44]. Importantly, dihydroartemisinin treatment at 5000 nM resulted in hematin–drug adducts widely associated with the parasite cell lysate but not detected in the Hz fraction [44].

After 32 h of artesunate exposure to parasites, Combrinck et al. [45] reported an increase in soluble heme and a decrease in the Hz content, two hallmarks of heme detoxification suppression. Of note, the observed effects of artesunate on the *Plasmodium* heme detoxification were less pronounced than the effects of chloroquine. Using this same method of heme and Hz quantification, Capci et al. [46] found that 24 h after oral treatment in *P. berghei*-infected mice, artesunate did not increase soluble heme or decrease the Hz content, while chloroquine at the same drug dosage did. However, artesunate reduced parasitemia and restored hemoglobin levels, while chloroquine did not. In common, both studies employed a prolonged period of drug exposure to the parasites and observed greater heme detoxification suppression for chloroquine than artesunate. Additional studies are needed, especially to compare short and long periods of treatment, to definitively and quantitatively elucidate the blocking of heme detoxification by endoperoxides.

## 3. Introduction to Redox Chemistry

One prevailing hypothesis is that after intense hemoglobin degradation, there is a peak of hematin biomineralization (Figure 2B) [22,45]; however, cytosolic heme remains practically unaltered as the parasite grows further [47]. In fact, Fe-PPIXs can play distinct roles in the *Plasmodium* biology as the parasite has evolved to control the levels and species of soluble Fe-PPIXs inside the DV for the following stage, which is the biomineralization of hematin. This is only possible because the DV membrane acts as a physiological barrier hampering the free flow of Fe-PPIXs.

One notable example of how Fe-PPIXs can play distinct roles is the fact that not only does ferrous heme not spontaneously dimerize into β-hematin crystal, but it actually suppresses crystal formation when assayed under low molecular oxygen. One plausible explanation for this is that heme binds to hematin through μ-oxo-dimeric speciation, which antagonizes the head-to-tail dimer required for the formation of β-hematin crystals. As heme antagonizes hematin, this constitutes a proof-of-concept that antioxidant treatment by chemically reducing ferric hematin into ferrous heme can kill parasites, albeit not in a drug concentration range that is usable therapeutically [48].

Similar to heme, other protoporphyrins IX (PPIX), such as the zinc protoporphyrin (Zn^II^-PPIX) and the tin protoporphyrin (Sn^IV^-PPIX), can also bind to soluble hematin and inhibit β-hematin crystal formation while also being adsorbed on growing crystal surfaces and suppressing their elongation [49,50,51]. However, Sn^IV^-PPIX is five times more potent as a β-hematin inhibitor than its counterpart Zn^II^-PPIX [51]. Interestingly, both Sn^IV^-PPIX and Fe^II^-PPIX are potent β-hematin inhibitors and coincidently contain metals with two main oxidation states (i.e., redox-active) whereas zinc is redox-inactive. Supporting this observation, it was recently demonstrated that the antiplasmodial activity of the redox-active Fe^III^-PPIX is abrogated by pre-exposure to the antioxidant (*N*-acetyl-cysteine), but the same abrogation was not observed for the redox-inactive Zn-PPIX [52]. Based on this, it is possible that in addition to inhibiting β-hematin formation, PPIX may have an effect on *Plasmodium* redox homeostasis.

As outlined above, pharmacological strategies to affect the redox balance of Fe^II^/Fe^III^-PPIX by favoring the chemical reduction of ferric hematin into ferrous heme using antioxidants or using oxidants and alkylating agents that can concomitantly oxidize and alkylate heme to hematin are formal possibilities for malaria chemotherapy. The latter, which involves antimalarial peroxides, is the most successful pharmacological intervention for malaria yet discovered. Nowadays, there is compelling evidence that peroxides such as artemisinin behave as oxidants in regard to the phenotypic response they induce in the parasite cells, causing the augmented production of reactive oxygen species that are cytotoxic for parasites [53,54]. However, the molecular basis responsible for triggering these oxidant effects observed for artemisinin and other endoperoxides remained unclear for many years.

Owing to the finding disclosed in 1991 that artemisinin is activated by heme in *Plasmodium* parasite cells [2], the scientific community turned its attention to the study of the chemistry underlying this activation. The importance of this issue is twofold: first, heme is responsible for drug activation; second, heme is preserved along multiple parasite stages (at least in the asexual blood stages). Most of the heme inside the DV is transient and is rapidly oxidized into hematin. However, outside the DV, *Plasmodium* can provide a low but sustained cellular concentration of heme. These levels stem from de novo biosynthesis and are located most likely in the mitochondrion and apicoplast organelles [55,56]. Regardless of the exact site where an antimalarial peroxide resides, inside or outside the DV, these drugs would certainly find an environment that is enriched in heme levels.

The structural and mechanistic aspects of the reductive activation of artemisinin by heme and its underlying mechanism for *Plasmodium* heme detoxification were elucidated after the publication of two seminal works in 2002 [4,57]. Meunier et al. [57] demonstrated the first synthesis of the hematin–artemisinin adduct and its chemical characterization by mass spectroscopy and that of the demetallated adduct by nuclear magnetic resonance (Figure 1B). They were able to show that the synthesis can occur under mild conditions and can have high yields, and that this adduct indeed results from a reduction of the peroxide bond. Chauhan et al. [4] also synthesized the hematin–artemisinin adduct and observed that iron from the resulting adduct is likely to be in a ferric state (iron III). These authors demonstrated that in a clear contrast to hematin, the synthetic hematin–artemisinin adduct does not spontaneously dimerize in solution but could block β-hematin formation. In the following years, this bioreductive activation was demonstrated to remain intact for all antimalarial endoperoxides [58], including artemisinin derivatives [5], as well as in the synthetic class of ozonide molecules, such as 1,2,4-trioxolanes and 1,2,4,5-tetraoxanes [59,60,61].

Naturally, the next step was to determine the real extent and relevance of this reductive activation pathway under biogenic conditions using parasite cells. Early in vivo observations using *P. yoelii nigeriensis*-infected mice [62] and later in vitro observations using the asexual blood stage of a *P. falciparum* cell culture [63] under treatment by endoperoxides showed the biogenic formation of hematin–drug adducts (Figure 3). More importantly, these studies showed that the abundance of adduct formation correlated with the level of parasite resistance to the peroxide drug; namely, parasites resistant to artemisinin produced a low amount of hematin–drug adducts. Conversely, suppressing the formation of hematin–drug adducts is one of the many mechanisms deployed by *Plasmodium* to overcome its susceptibility to antimalarial peroxides.

Subsequently, it became clear that heme is the drug-activating agent and that antimalarial peroxides do alkylate heme. However, for many years, the general idea was that the mechanism for the antiplasmodial activity of peroxide drugs was radical-induced protein alkylation, considered to be the reason for the inactivation of a variety of biomacromolecules essential for parasite growth and survival [64]. It was assumed that the alkylation of heme towards hematin–drug adducts was solely a step for generating end products of a typical bioreductive activation process. The concept that hematin–drug adducts are benign or harmless species to *Plasmodium* parasites started to change after mounting evidence indicated that these adducts possesses reactivity and pleiotropic effects towards *Plasmodium*.

In 2007, a synthetic hematin–artemisinin adduct was shown to block β-hematin crystal formation when added to the reaction at a 5:1 ratio relative to hematin, being three times more potent than the reference inhibitor chloroquine, while none of the tested antimalarial endoperoxides (artemisinin, artesunate, and artemether) were able to block β-hematin formation [65]. This finding was consistent with the vast literature denoting the potential of metalloporphyrins antagonizing hematin biomineralization [49,66]. However, most metalloporphyrins typically present weak antimalarial activity [50,51,67]; this is assumed to be because the DV membrane hampers metalloporphyrin flow into and out of DV [51].

The precise mechanism of how hematin–artemisinin adducts can inhibit β-hematin formation and more importantly, how this translates to antiparasitic activity, have remained unclear until recent years, when finally Ma et al. [41] demonstrated that synthetic hematin–drug adducts produced from artesunate or artemisinin can adsorb at the kinks in the β-hematin crystal surfaces. This site on the crystal surface is by far the least effective for intervention, but this weakness is compensated by the fact that these adducts inhibit crystal growth in an irreversible manner, which to date has only been reproduced for metalloporphyrins. These hematin–drug adducts were endowed with in vitro antimalarial activity against drug-sensitive and drug-resistant strains at the *P. falciparum* asexual blood stages, being as potent as artesunate or artemisinin. For instance, the fifty percent inhibitory concentration (IC_50_) value for hematin is 24,000 ± 4000 nM [52], while the IC_50_ values for hematin–drug adducts can be as low as 10 nM. Finally, these authors demonstrated by mass spectroscopy analysis that the hematin–drug adducts but not their parent drugs (artesunate or artemisinin) were bound to Hz isolated from parasite samples, suggesting that the observed β-hematin inhibition using cell-free conditions translated into a blockade of *Plasmodium* heme detoxification in a cell culture (Figure 4).

## 4. Interplay between Redox Homeostasis and Heme Detoxification

Inside the DV, heme detoxification is centered on the conversion of soluble hematin into an insoluble head-to-tail hematin dimer; in other words, it is centered on suppressing soluble Fe-PPIX species. Outside the DV, *Plasmodium* synthesizes heme and expresses endogenous reductants to tightly regulate iron redox homeostasis to prevent the oxidation of heme to hematin. The findings disclosed by Ma et al. [41] provided the first answers to the question as to whether the products of redox chemistry between heme and artemisinin—hematin–drug adducts—have strong antimalarial activity by blocking *Plasmodium* heme detoxification. This could be interpreted as an interplay between iron redox homeostasis and heme detoxification. Specifically, the redox chemistry of heme and Artemisinin results in the formation of an endogenous heme detoxification suppressor. Supporting this are the following observations: (i) redox chemistry between heme and peroxide drugs is mandatory and conserved for all drugs; (ii) redox chemistry is present in all three stages of the asexual blood life cycle (rings, trophozoites, schizonts); (iii) the resulting hematin–drug adducts can bind to hematin and inhibit β-hematin crystal formation.

Despite all this, a persistent question was whether antimalarial endoperoxides display β-hematin inhibitory activity when ferrous heme is employed as a starting reactant, reproducing a cellular environment in which a peroxide drug encounters heme and further proceeds in the heme detoxification process to suppress Hz formation. Recently, Ribbiso et al. [40] addressed this by making use of an in situ glutathione-mediated reduction of ferric hematin into ferrous heme, subsequently incubated with drugs as a pre-step for β-hematin formation. Dihydroartemisinin, artemether, and artesunate displayed concentration-dependent β-hematin inhibitory activity when assayed on heme but not on hematin, being as potent as or even more potent than reference 4-aminoquinolines, chloroquine, and amodiaquine. The authors argued that endoperoxides alkylated heme and this alkylation produced adducts capable of blocking Hz formation and consequently suppressing heme detoxification (Figure 5). Of note, this method is particularly interesting because even if an inhibitor that does not bind to ferrous heme in the pre-step is screened, once an acid buffer is added, oxidation of heme into hematin occurs, allowing the hematin formed to interact with an inhibitor with affinity for hematin and thus to display inhibitory activity.

Regarding the three steps involved in the formation of β-hematin crystals when assayed using ferrous heme—i.e., oxidation of heme into hematin, hematin dimerization, and crystal formation—the findings indicate that when endoperoxides undergo hematin–drug adduct formation, they have an inhibitory effect on all three steps of hematin biomineralization. Firstly, adduct formation itself consumes heme, which would naturally be converted into hematin [40]. Secondly, the resulting hematin–drug adducts can bind to non-crystalline and pre-crystalline forms (monomers and dimers) within β-hematin formation. The resulting potency of artemisinins to inhibit β-hematin formation assayed against ferrous heme may be interpreted as evidence of artemisinin-hematin adducts binding to hematin since potency to inhibit β-hematin crystals could not be merely achieved by the consummation of heme as a solute in the reaction. This is further supported by the fact that a synthetic hematin–artemisinin adduct can potently inhibit β-hematin formation when assayed using ferric hematin, highlighting the affinity of adducts for the monomers and dimers of hematin [65]. Finally, hematin–drug adducts can be adsorbed on the surface of growing crystals by irreversibly blocking their elongation [41] (Figure 5).

By correlating the compelling findings of Ribbiso et al. [40] and Ma et al. [41], it was observed that all three tested endoperoxides (dihydroartemisinin, artemether, and artesunate) presented almost identical IC_50_ values for the β-hematin inhibitory activity assayed on ferrous heme, although dihydroartemisinin was found to be a more potent antimalarial agent than artemether or artesunate [68]. A similar trend was also observed for hematin–drug adducts since hematin–drug adducts produced from either artemisinin or artesunate were equipotent in inhibiting β-hematin crystal growth and inhibiting parasite growth. In contrast, amodiaquine has been found to be a more potent β-hematin inhibitor than chloroquine in both ferrous and ferric assays. In fact, amodiaquine is a more potent antimalarial agent than chloroquine [37,68]. For 4-aminoquinolines, a correlation between β-hematin inhibitory activity and antiparasitic activity is observed; that is, the most potent β-hematin inhibitor is also the most potent antimalarial in a cell culture. This kind of correlation is not clearly observed for either endoperoxides inhibiting β-hematin or hematin–drug adducts binding to β-hematin crystals and killing parasites. This apparent absence in the structure –activity relationship may be interpreted as the Fe-PPIX component of hematin–drug adducts being the main structural contributor for β-hematin inhibition; perhaps the Fe-PPIX component is the sole contributor. Alternatively, it is possible that all hematin–drug adducts have a similar drug speciation and lipophilicity profile, which contrasts with the parental drugs [69,70,71]. For instance, among the three main endoperoxides (dihydroartemisinin, artemisinin, and artesunate), dihydroartemisinin has the lowest chemical stability [70], in part because of its likely pH-dependent metabolization into a monoketo-aldehyde-peroxyhemiacetal species [71], which may contribute to its antimalarial profile [72]. Artesunate, for instance, typically displays superior antimalarial activity to artemisinin, which is attributed to the side chain attached at the C-10 position enhancing its lipophilicity [72,73,74]

## 5. Heme Alkylation versus Protein Alkylation: The Debate

The main hypothesis proposed for the antimalarial activity of peroxides is the radical-induced alkylation of client molecules [10]. These may include heme, peptides, proteins, lipids, and others. A prevailing hypothesis is that a stochastic process is accomplished for radical-induced alkylation [64,75]. The alkylation and inactivation of essential proteins and lipids can certainly have an impact on parasite viability and growth. Regarding heme, its alkylation can cause an imbalance in the heme/hematin ratio, therefore altering iron redox homeostasis. The product of this alkylation is also an endogenous and potent antimalarial agent [41]. Via this reasoning, the well-established capability of hematin–drug adducts to inhibit β-hematin crystal formation and the recent observation that these adducts present in vitro antimalarial activity against the asexual blood stages of *P. falciparum* with potency comparable to parental drugs [41] can be interpreted as the alkylation of heme by endoperoxides involved in antimalarial activity (Figure 6).

Unlike the parent peroxides, the ability of hematin–drug adducts to alkylate molecules is jeopardized, especially considering that the peroxide bond is no longer available to participate in the Fenton reaction, which is needed for radical formation. The structural necessity of the endoperoxide bond for artemisinin to display antimalarial activity has been directly assessed by multiple studies from different research groups [76,77]. Further evidence for this was recently disclosed by systematically showing that the chemical replacement of endoperoxide with an ether bond is deleterious for the antimalarial activity of artemisinin derivatives [78]. With the ability to alkylate molecules compromised, how can hematin–drug adducts kill *Plasmodium* parasites?

As mentioned above, the two main hallmarks of heme detoxification suppression are the augmentation of free Fe-PPIX levels and the reduction of the Hz content. Compelling studies have showed that hematin–drug adducts inhibit β-hematin growth and Hz formation [4,41,65]; however, increased free Fe-PPIX levels in parasites treated with these drugs have not been demonstrated. Regardless, considering the inability of this class of adducts to alkylate molecules unlike their parent drugs, in addition to the suppressive effects of this class of drug on heme detoxification, to date, the most plausible mechanism is that these drugs suppress *Plasmodium* heme detoxification. It is also possible that these hematin–drug adducts, viewed as modified Fe-PPIX, may reproduce the toxic role of free Fe-PPIX in killing parasites, as typically observed in parasites treated with classic heme detoxification suppressors such as chloroquine and amodiaquine [52]. Figure 7 illustrates a pharmacophore model that captures the essence of Fe-PPIX as a structural determinant for Hz recognition and the sesquiterpene lactone from artemisinins as a determinant for cell permeability and drug transportation inside the DV.

## 6. Frontier Questions and Perspectives

A full understanding of the cellular distribution and accumulation of hematin–drug adducts remains an unmet challenge; as such, identifying these characteristics could provide a model to explain the scope and limitation of this class of potential antimalarial therapeutics. Via this reasoning, the parasite cytosol is enriched in reductants such as glutathione [79]. This kind of environment may reduce the iron atom of hematin–drug adducts from a ferric to a ferrous state and this could provide a pathway for a hypothetical parasite-killing mechanism mediated by redox-activity. To date, however, the oxidative behavior of iron in these adducts under biogenic conditions is not fully understood, nor is it clear whether a toggle between ferrous and ferric states exists. Furthermore, while different research groups have synthesized, isolated, and chemically characterized hematin–drug adducts [4,5,41,57], their stability and chemical speciation in solution remain poorly understood. This is important in view of the fact that experiments involving the titration of heme by endoperoxides have indicated the existence of chemical transformation and degradation in addition to heme alkylation. For example, Zhang and Gerhard [9] observed, by UV-vis spectroscopy, that the reaction between heme and artemisinin in dimethyl sulfoxide behaves as a first-order reaction, with the instantaneous formation of hematin–artemisinin adducts. However, the signal of the Soret band for Fe-PPIX declined over the time, where the authors estimated a half-life of 25.5 min. Very similar results were found by Giannangelo et al. [80] when monitoring the redox chemical reaction between heme and the synthetic 1,2,4-trioxolane arterolane (Figure 7). Heller et al. [81] monitored this kind of reaction with artesunate by nuclear magnetic resonance in an aqueous medium and observed that it occurred more slowly than anticipated, without the complete consummation of reactants even after 3 h of incubation. These different outcomes may be due to the experimental conditions, such as the solvent, temperature, and apparent pH. Clearly, much more work remains to be done to better understand this redox reaction.

Elucidating the action mechanisms of antimalarial drugs could accelerate the development of new and effective malaria therapies. Several studies have emphasized the essential nature of the relationship between peroxide drugs and heme. It is now clear that this relationship has deep ties with the two main aspects of *Plasmodium* biology: redox homeostasis and heme detoxification. Integrating the concepts of how antimalarial peroxides can interfere in these processes is of pivotal importance. Despite the progress made, as revised above, there still questions to be answered before a pharmacological model can be developed that is capable of reconciling most of these processes.

One outstanding question is why antimalarial endoperoxides, such as artemisinin, do not potently inhibit the *Plasmodium* heme detoxification. For instance, the cellular concentration of the biogenic hematin-dihydroartemisinin adduct in *P. falciparum* can range from 1000 to a 25,000 nM depending on the parasite stage in a *P. falciparum* culture treated with a dihydroartemisinin bolus of 5000 nM [44]. This concentration is far higher than the concentration of approximately 3 nM employed by Combrinck et al. [45] to demonstrate that artesunate treatment can cause augmentated free heme levels and a decline in the Hz content. Despite this, the effects of artesunate on *Plasmodium* heme detoxification are less pronounced than those of chloroquine treatment.

A second riddle is this: if an antimalarial peroxide alkylates heme by forming species that have a high affinity for binding to hematin and thus are highly effective in suppressing heme detoxification, a peroxide drug would be expected to exhibit appreciable accumulation inside the DV in order to get in closer proximity to the client molecule. Early studies have indicated that artemisinin and its chemical variants conjugated with fluorescent tags accumulate in the cytosol and to a lesser extent, in the DV [59,82]. However, a more recent study using a fluorescent artemisinin observed drug accumulation in the cytosol and no detectable accumulation in the DV [83]. The full extent of peroxide drug accumulation inside the DV has yet to be elucidated; nonetheless, it is notable that these drugs have no preferential accumulation for the DV, in contrast to the compelling observations of antimalarial 4-aminoquinolines, which are strong heme detoxification suppressors, accumulating in the DV [84,85].

Finally, it is important to place the pharmacological models in Figure 6 and Figure 7 in the context of the asexual blood stages of *Plasmodium*, where direct evidence of the bioreductive activation of artemisinins by heme was observed. For the parasites in the sexual (gametocytes) and the hepatic (sporozoites) stages, the real extent of this bioreductive activation remains elusive. This is because currently there is no direct evidence of bioreductive activation for these parasites. It is known that these parasites have heme, which is supplied by de novo biosynthesis [20,55,56]. Presumably, if these parasite stages do have heme, bioreductive activation can take place; however, interpretation is not as simple as implied. Bioavailability of heme in these parasite stages has been less studied [55,56], and susceptibility of these parasites to artemisinin treatment can vary [68,86]. For instance, the asexual blood stages and the young gametocytes are quite susceptible to dihydroartemisinin by typically displaying IC_50_ values in a low nanomolar range; this contrasts with mature gametocytes, which are approximately two hundred times less susceptible to dihydroartemisinin treatment [87]. This is intriguing, especially considering that the mature gametocytes are quite susceptible to methylene blue, a drug whose mechanism of action is attributed to oxidative stress and a redox imbalance [88], resembling the mechanisms of artemisinins. In principle, the different outcomes in parasite susceptibility for methylene blue and artemisinins may be due to a dissimilarity in the molecular basis responsible for triggering oxidative stress [89]. Clearly, experimental quantification of bioreductive activation for these parasite stages is required to answer this frontier question.

## 7. Conclusions

*Plasmodium* operates a synchronized system to mitigate a free flow of Fe-PPIXs across the DV membrane. Multiple studies indicate that antimalarial endoperoxides, such as 1,2,4-trioxane artemisinin and 1,2,4-trioxolane arterolane, operate a mechanism of radical-induced alkylation mediated by heme. A plethora of molecules can be alkylated, and a plausible pharmacological model is that the antimalarial activity of endoperoxides can be achieved by alkylating heme as well. In support of this notion is the fact that hematin–drug adducts are highly abundant, strong antimalarials, and potent antagonists of hematin biomineralization. These alkylated heme species recognize soluble and insoluble hematin pools, but as modified Fe-PPIX species, they do not mineralize into Hz crystals but, rather, block hematin biomineralization. Furthermore, heme alkylation alters the heme/hematin ratio and can thereby induce an imbalance in the redox homeostasis of iron species.

Similar to other metalloporphyrins, hematin–artemisinin adducts bind irreversibly to growing β-hematin crystals, and this initiates the suppression of heme detoxification. These adducts act as a modified Fe-PPIX structure containing two pharmacophores: Fe-PPIX recognizes and inhibits β-hematin formation while the highly lipophilic sesquiterpene lactone from artemisinin contributes to overcome the poor lipophilicity of metalloporphyrin, in addition to its plausible interaction with parasite targets. Via this reasoning, hematin–artemisinin adducts could be seen as being heterobivalent with respect to their recognition of multiple molecular targets. Heterobivalency is one of the principles of the molecular hybridization approach for designing hybrid-based drugs [90], which reinforces the notion of Fe-PPIX as an antimalarial pharmacophore component, and adds a new potential rationale for hybrid-based drug design.

## Figures and Tables

**Figure 1 pharmaceuticals-15-00060-f001:**
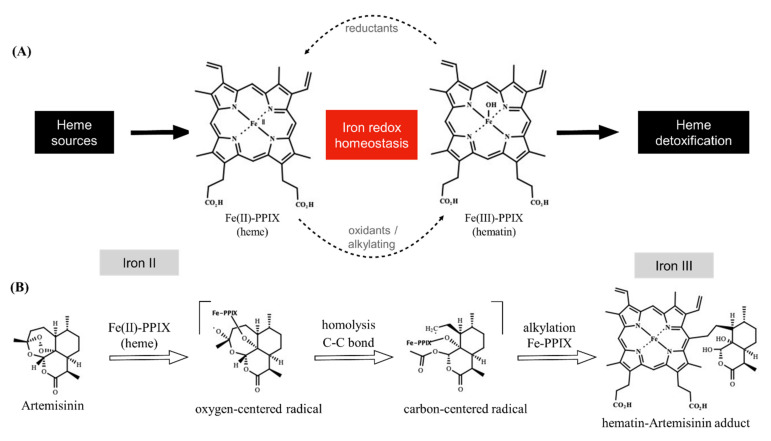
Overview of iron redox chemistry and the drug blockage. Panel (**A**) shows that the oxidative state of iron is regulated in the parasite. Heme is oxidized by oxygen into hematin, which is then biomineralized within the heme detoxification process. A shift in the homeostasis towards heme can be induced by endogenous reductants (a formal possibility being glutathione). A shift in the homeostasis towards hematin can be induced by alkylating agents, such as artemisinin. Panel (**B**) shows that artemisinin undergoes reductive activation by heme, producing a radical species. This can alkylate a variety of molecules (not depicted), including Fe-PPIX. Heme alkylation results in the hematin–artemisinin adduct (for clarity, only one isomer adduct is depicted). Note that heme alkylation is accompanied by the exchange of the iron oxidative state from ferrous to ferric Fe-PPIX. Unless specified, iron is in a ferric state. Fe-PPIX = iron protoporphyrin IX.

**Figure 2 pharmaceuticals-15-00060-f002:**
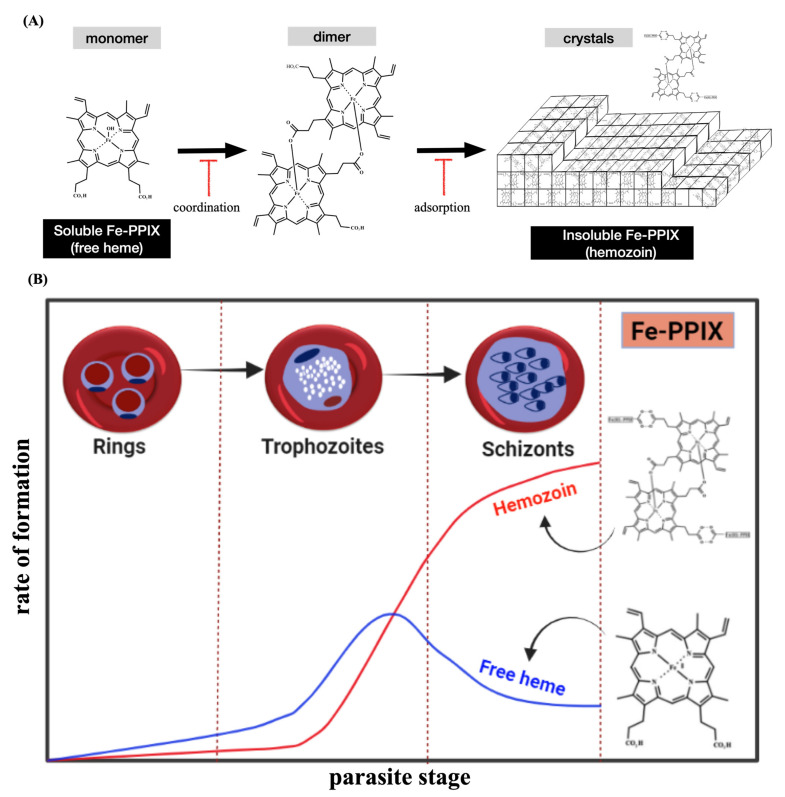
Overview of the heme detoxification process. Panel (**A**) shows that this process is centered on the conversion of soluble Fe-PPIX into insoluble Hz crystals. Antimalarials can typically block this process by binding, via iron coordination chemistry, to soluble Fe-PPIX and by capping the surface of Hz crystals via adsorption mechanisms. Panel (**B**) shows that the phenotype of heme detoxification within the asexual blood stage of *Plasmodium*, characterized by a decline in the levels of soluble heme (free heme) and a sharp rise in Hz formation.

**Figure 3 pharmaceuticals-15-00060-f003:**
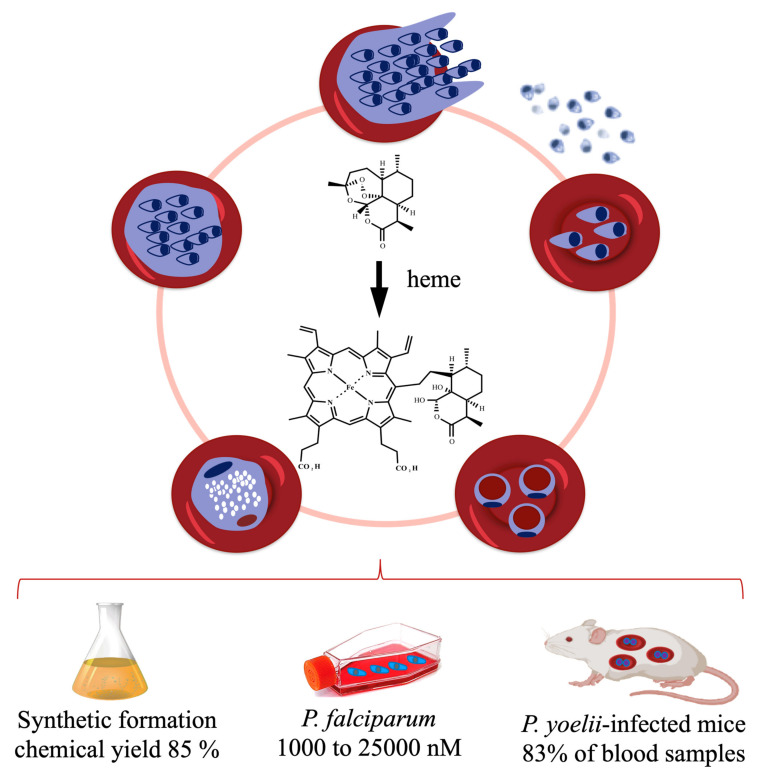
Activation of antimalarial endoperoxides by heme to hematin–drug adducts. This can be synthetically reproduced in the laboratory under mild conditions, producing high chemical yields [57]. Biogenically formed hematin–drug adducts were detected in high concentrations in a cell culture of *P. falciparum* treated with 5000 nM of dihydroartemisinin [62] and were detected at 83% in blood samples from infected mice treated with artemisinin [63]. For *P. falciparum*, a hematin-dihydroartemisinin adduct is produced, but for clarity its structure is not depicted here.

**Figure 4 pharmaceuticals-15-00060-f004:**
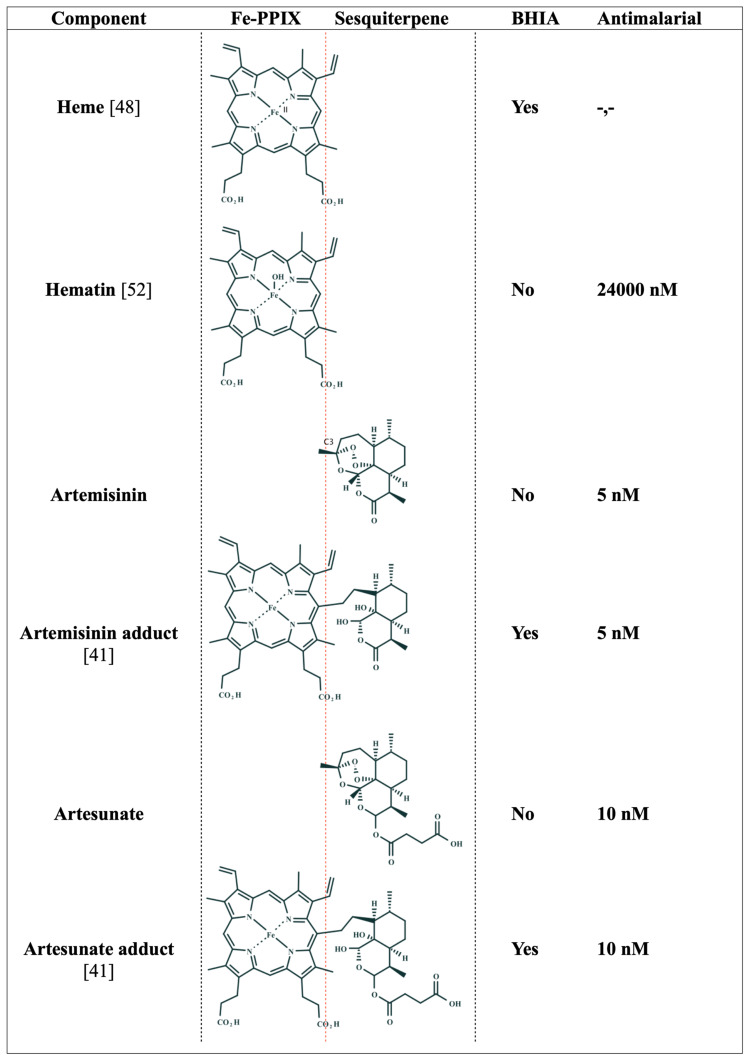
Structure–activity relationship of Fe-PPIX, sesquiterpenes, and their hematin–drug adducts. BHIA = β-hematin inhibitory activity. Yes indicates inhibitory activity; No indicates the absence of inhibitory activity. Antimalarial = IC_50_ values of the in vitro antiparasitic activity against the asexual blood stages of *P. falciparum*. Adducts are drawn as deacetylated structures (indicated by C-3 carbon) according to reference [63].

**Figure 5 pharmaceuticals-15-00060-f005:**
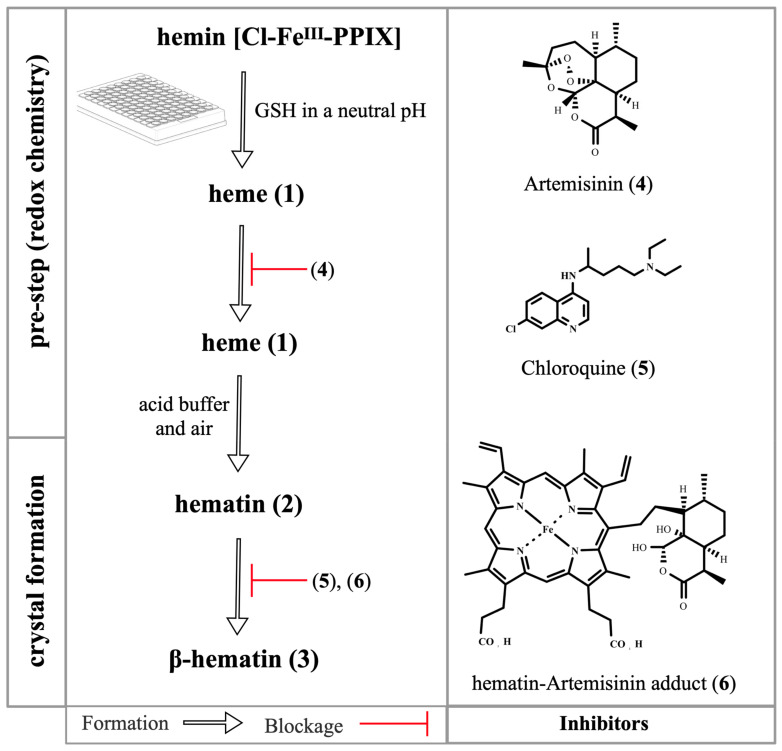
Overview of the β-hematin crystal formation and drug blockage. A typical assay employs hematin (**3**) as a starting reactant, but to assess the redox chemistry, a pre-step is employed, where hemin chloride is reduced by glutathione (GSH) into heme (**1**). Upon addition of an acid buffer and exposure to air, heme (**1**) is oxidized back to hematin (**2**) and is gradually dimerized into β-hematin dimers (**3**) to form crystals (not depicted here) [40,48]. The artemisinin (**4**) alkylates the heme (**1**) and the hematin–artemisinin adduct (**6**) is formed. The adduct (**6**) and 4-aminoquinolines in general, exemplified here by chloroquine (**5**), bind to hematin (**2**) and inhibit its dimerization (**3**).

**Figure 6 pharmaceuticals-15-00060-f006:**
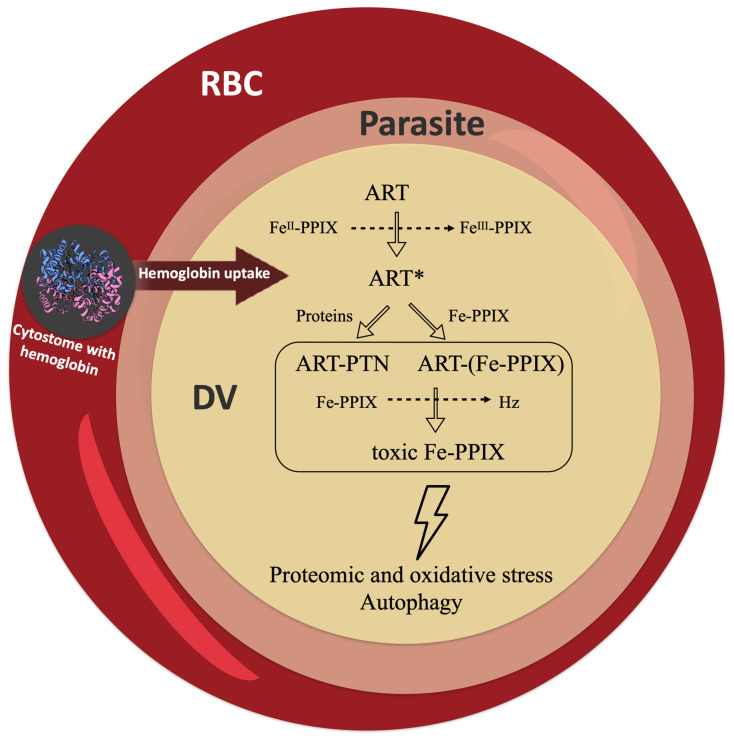
Potential pathways for the radical-induced alkylation of proteins and heme. Artemisinin (ART) is activated by heme into a radical species (ART*). This can alkylate client molecules, such as proteins (PTN) and iron protoporphyrin IX (Fe-PPIX). The alkylation of proteins (ART-PTN) can have an impact on the pathways of protein degradation by causing proteomic stress, autophagy, and other effects. The alkylation of heme [ART-(Fe-PPIX)] can suppress heme detoxification, leading to a buildup of toxic Fe-PPIX and causing oxidative stress, autophagy, and other effects. RBC = red blood cell; DV = digestive vacuole.

**Figure 7 pharmaceuticals-15-00060-f007:**
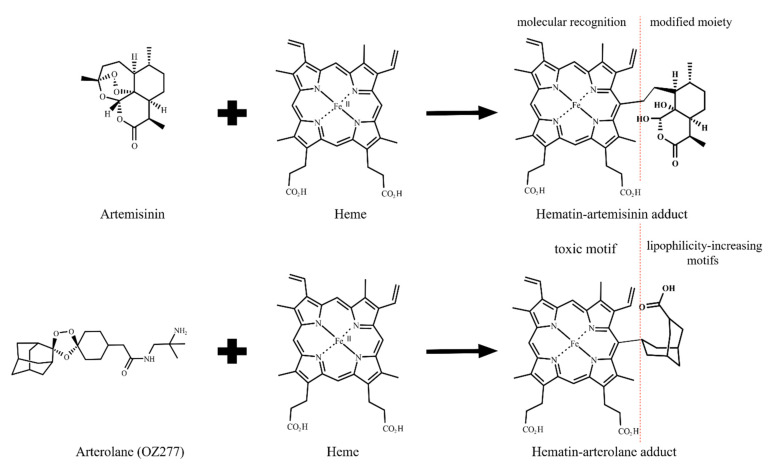
Formation of hematin–drug adducts from artemisinin and arterolane. Representation of the two possible pharmacophore domains: porphyrin and aliphatic groups. The adducts are embodied into the heme detoxification process by structural complementarity to porphyrin. However, the modified moieties impair a regular heme detoxification process, causing the suppression of heme detoxification. The modified moieties are aliphatic groups and are likely to be especially relevant for cell permeability and drug transportation for inside the DV.

## Data Availability

Not applicable.

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
