# Peer review of "The Role of the Iron Protoporphyrins Heme and Hematin in the Antimalarial Activity of Endoperoxide Drugs"

_pharmaceuticals, 2022, doi:10.3390/ph15010060_

Round 1

Reviewer 1 Report

Authors describe role of iron as bioreductive acticvactors of artemisinins and endoperoxide drugs.

Review is well organized but NOT consider other hypothesis that correctly MUST be added to complete view to readers.

So suggest to read and include these references in description of your paper:

Richard K. Haynes, Kwan-Wing Cheu, David N'Da, Paolo Coghi Monti. Some Current Considerations on the Mechanism of action of Artemisinin Antimalarials : Part 1 – The ‘Carbon Radical’ and ‘Heme’ Hypotheses,Infectious Disorders – Drug Targets, 2013, 13, 217-277

P. Coghi, N. Basilico, D. Taramelli, W. Chan, R.K. Haynes, D.Monti.Interaction of Artemisinins with Oxyhemoglobin Hb-FeII, Hb-FeII, CarboxyHb-FeII, Heme-FeII, and Carboxyheme FeII: Significance for Mode of Action and Implications for Therapy of Cerebral Malaria. ChemMedChem 2009, 4, 12, 2045-2053.

Hence, this reviewer indicate accept this MS for publication after major revision in Molecules.   

  

  

Author Response

Reviewer #1 Authors describe role of iron as bioreductive activators of artemisinins and endoperoxide drugs. Review is well organized but NOT consider other hypotheses that correctly MUST be added to complete view to readers.

Authors reply: We thank the reviewer for the careful review, for the important suggestions to improve the manuscript and for appreciating the interest and quality of our work. We agree with the reviewer that a consideration of other hypothesis than bioreductive activators is critical for this review paper. In this version of the manuscript, we have included a discussion of other hypotheses and mechanisms.

Reviewer #1 So suggest to read and include these references in description of your paper:

Richard K. Haynes, Kwan-Wing Cheu, David N'Da, Paolo Coghi Monti. Some Current Considerations on the Mechanism of action of Artemisinin Antimalarials : Part 1 – The ‘Carbon Radical’ and ‘Heme’ Hypotheses,Infectious Disorders – Drug Targets, 2013, 13, 217-277. P. Coghi, N. Basilico, D. Taramelli, W. Chan, R.K. Haynes, D.Monti.Interaction of Artemisinins with Oxyhemoglobin Hb-FeII, Hb-FeII, CarboxyHb-FeII, Heme-FeII, and Carboxyheme FeII: Significance for Mode of Action and Implications for Therapy of Cerebral Malaria. ChemMedChem 2009, 4, 12, 2045-2053. Hence, this reviewer indicates accept this MS for publication after major revision in Pharmaceuticals.   

Authors reply: We thank the reviewer for providing these valuable references. The first study led by researchers at Università degli Studi di Milano (Italy) and the second one, led by scientists at The Hong Kong University of Science and Technology (China) are interrelated to the subject of this review paper. The content of these papers (references 38, 69 and 73) was discussed in this new manuscript version.

Reviewer 2 Report

The authors have aimed to produce a review that covers a wide breadth of mechanistic topics around redox chemistry, heme/hematin and antimalarial chemotherapies. This review would be a useful resource for the field, given the importance of endoperoxide drugs in the current frontline therapies for malaria treatment. However, there are two major faults of this current draft of the manuscript. First, the English is problematic throughout the document, often obscuring the intended meaning of the text. The manuscript would benefit greatly from a more thorough general review of the language for increased clarity, reduction in repetition, and improved grammar and sentence structure. Second, there are unproofread sections where the manuscript is missing information or has the incorrect citation associated with a sentence. Some examples include the legend for figure two does not describe panels B and C of the figure, and reference 55 is incorrect. Third, the manuscript would benefit greatly from an increased focus on gaps in knowledge of this field. There is good discussion of this immediately before the conclusions, but perhaps putting the authors’ perspectives on important outstanding questions in an independent section and expanding on them would provide the readers with a better idea of the future of this field.

Below are detailed some specific suggested changes. Careful reading of the manuscript for sentence clarity ended after approximately Line 200, but clarity issues are abundant throughout the document.

Line 53: The sentence beginning on this line needs editing for clarity

Line 56:  The sentence beginning on this line needs editing for clarity

Line 61: This isn’t a complete sentence

Line 68: “there” should be “there is”

Line 79: What is meant by “Like any other unicellular pathogen?” Not all unicellular pathogens reside in a heme-rich RBC environment.

Line 81: This sentence needs editing for clarity

Line 90: This sentence needs commas to separate members of the list

Line 92: This sentence needs editing for clarity

Line 98: Figure 2 should be referenced in this paragraph

Line 101: This sentence needs editing for clarity

Line 110: This sentence needs editing for clarity

Line 115: This sentence needs editing for clarity

Figure 2: the inhibitors “drug coordination” and “drug adsorption” should be at the base of the inhibition “T” symbol as is common practice

Figure 2: Panels B and C don’t add to the review as they are. Could consider omitting

Line 131: The legend here doesn’t mention panels B or C at all. These should be described or omitted. They are also not referenced in the text.

Line 132: “Scheme 36” is not defined anywhere and isn’t obvious the meaning

Line 142: This paragraph needs editing for clarity

Line 173: This sentence needs editing for clarity

Line 211: This sentence needs editing for clarity

*English issues (syntax and word choice) is detrimental to the manuscript clarity throughout. This Reviewer discontinued marking clarity issues after this point.

Line 218: Reference 55 is incorrect for this sentence. All references should be double checked before resubmission

Figure 3: This figure is not integrated into the text well and does not benefit the reader sufficiently in its current state. It would benefit from a more thorough explanation in the text or in the figure legend or from a consideration of omission. Are the red arrows meant to show where in the parasite cycle the adducts were detected?

Line 263:

Figure 4: Formatting issue precludes adequate evaluation by this reviewer

Figure 5: Similar to Figure 2, inhibitors should be at the base of the inhibition “T” symbol as is common practice.

Figure 5: There is no specified inhibitor associated with the inhibition “T” symbol between Hematin (2) and beta-Hematin (3)

Figure 7: There should be “+” signs between the reactants in each reaction

Author Response

Reviewer #2: The authors have aimed to produce a review that covers a wide breadth of mechanistic topics around redox chemistry, heme/hematin and antimalarial chemotherapies. This review would be a useful resource for the field, given the importance of endoperoxide drugs in the current frontline therapies for malaria treatment.

Authors reply: We thank the reviewer for the careful review and for recognizing the importance of this line of research.

Reviewer #2: However, there are two major faults of this current draft of the manuscript. First, the English is problematic throughout the document, often obscuring the intended meaning of the text. The manuscript would benefit greatly from a more thorough general review of the language for increased clarity, reduction in repetition, and improved grammar and sentence structure.

Authors reply: From the reviewers’ comments we feel that we did not write our manuscript with sufficient clarity. We fully agree with the reviewer that, in the first version of the manuscript, there was a lack in clarity. The authors have revised the manuscript with one another and then asked the assistance of an expert in the English grammar. We hope that the reviewer agree that this version is now suitable for publication.

Reviewer #2: Second, there are unproofread sections where the manuscript is missing information or has the incorrect citation associated with a sentence. Some examples include the legend for figure two does not describe panels B and C of the figure, and reference 55 is incorrect.

Authors reply: The missing information occurred during the conversion of Microsoft Word file into the Pharmaceuticals template. This conversion was performed by the MDPI press and we were not aware of this until now. Regardless, we have now uploaded the manuscript version where figures were saved in a format file to avoid any kind of mistake. Regarding reference 55, this is our fault, which is now correct in this version.

Reviewer #2: Third, the manuscript would benefit greatly from an increased focus on gaps in knowledge of this field. There is good discussion of this immediately before the conclusions, but perhaps putting the authors’ perspectives on important outstanding questions in an independent section and expanding on them would provide the readers with a better idea of the future of this field.

Authors reply: We thank the reviewer for the important suggestions to improve the manuscript and for appreciating the interest and quality of our work. We agree with the reviewer that a consideration of perspectives in the field of mechanism of action of antimalarial peroxides is critical for this review paper. In this version of the manuscript, we made a new section that includes remaining questions and perspectives.

 Reviewer #2: Below are detailed some specific suggested changes. Careful reading of the manuscript for sentence clarity ended after approximately Line 200, but clarity issues are abundant throughout the document.

Line 53: The sentence beginning on this line needs editing for clarity

Line 56:  The sentence beginning on this line needs editing for clarity

Line 61: This isn’t a complete sentence

Line 68: “there” should be “there is”

Line 79: What is meant by “Like any other unicellular pathogen?” Not all unicellular pathogens reside in a heme-rich RBC environment.

Line 81: This sentence needs editing for clarity

Line 90: This sentence needs commas to separate members of the list

Line 92: This sentence needs editing for clarity

Line 98: Figure 2 should be referenced in this paragraph

Line 101: This sentence needs editing for clarity

Line 110: This sentence needs editing for clarity

Line 115: This sentence needs editing for clarity

Figure 2: the inhibitors “drug coordination” and “drug adsorption” should be at the base of the inhibition “T” symbol as is common practice

Figure 2: Panels B and C don’t add to the review as they are. Could consider omitting

Line 131: The legend here doesn’t mention panels B or C at all. These should be described or omitted. They are also not referenced in the text.

Line 132: “Scheme 36” is not defined anywhere and isn’t obvious the meaning

Line 142: This paragraph needs editing for clarity

Line 173: This sentence needs editing for clarity

Line 211: This sentence needs editing for clarity

*English issues (syntax and word choice) is detrimental to the manuscript clarity throughout. This Reviewer discontinued marking clarity issues after this point.

Line 218: Reference 55 is incorrect for this sentence. All references should be double checked before resubmission

Figure 3: This figure is not integrated into the text well and does not benefit the reader sufficiently in its current state. It would benefit from a more thorough explanation in the text or in the figure legend or from a consideration of omission. Are the red arrows meant to show where in the parasite cycle the adducts were detected?

Line 263:

Figure 4: Formatting issue precludes adequate evaluation by this reviewer

Figure 5: Similar to Figure 2, inhibitors should be at the base of the inhibition “T” symbol as is common practice.

Figure 5: There is no specified inhibitor associated with the inhibition “T” symbol between Hematin (2) and beta-Hematin (3)

Figure 7: There should be “+” signs between the reactants in each reaction

Authors reply: We have revised the main text and figures accordingly to reviewer’s comments. Regarding Figure 3, we agree with the reviewer that there was no integration between text and figure, but we have now integrated them. We are aware that this figure is more elusive than explanatory, but we feel this is important to provide readers an easy notion of this line of research, especially for readers without a previous background in the subject. We feel this figure has a potential for educational purposes.

Round 2

Reviewer 1 Report

the authors have improved paper as suggested

but I invited authors to add 2 references !

In the paper still lack one reference important :

Richard K. Haynes, Kwan-Wing Cheu, David N'Da, Paolo Coghi Monti. Some Current Considerations on the Mechanism of action of Artemisinin Antimalarials : Part 1 – The ‘Carbon Radical’ and ‘Heme’ Hypotheses,Infectious Disorders – Drug Targets, 2013, 13, 217-277.

So pls ADD  this reference and comment into to your manuscript ( maybe in section 6 when describe methylene blue...)

Hence this referee indicate accept this paper after major revision in pharmaceuticals

Author Response

Authors reply: We thank the reviewer for the careful review, for the important suggestions to improve the manuscript and for appreciating the interest and quality of our work. From the reviewers’ comments we feel that we did not revise the reference section with enough clarity. In this version of the manuscript, we have correctly included the reference cited above as number [89]. We hope that the reviewer agree that this version is now suitable for publication.

Reviewer 2 Report

The authors have put together a comprehensive, informative and thought-provoking manuscript. The field of malaria drug discovery will benefit from this paper by it providing a consolidated understanding of the relationships between iron the protoporphyrins heme and hematin, and antimalarials. This includes the importance of heme/hematin-drug adducts for comparatively actuating antimalarial activity. The end of the review posits a number of meaningful questions that may guide future studies regarding the mechanisms of activity of hematin-drug adducts and the potential use/directed development of such adducts for future antimalarial therapy. 

Below are specific minor suggestions and comments: 

Fig 2C: How do the hemozoin levels reduce in panel during the schizonts stage? Do antimalarials break apart hemozoin dimers in addition to preventing formation? 

Fig 2B and C: Label the Y axes of Fig 2B and C. Do the red and blue lines represent hemozoin and free heme production rate or the intracellular concentrations of each molecule?

Line 222: It would be helpful to relate this dynamic to Figures 2B and C. 

Author Response

Authors reply: Once again, we thank the reviewer for the careful review, for the important suggestions to improve the manuscript and for appreciating the interest and quality of our work. From the reviewers’ comments we feel that we did not design Figure 2 with sufficient clarity. We fully agree with the reviewer that, in the prior version of the manuscript, there was a misleading concept in panel C. The reviewer is right about the effect of drugs in the Hz content; namely, drug do prevent Hz formation instead of to break apart the crystal components. We have now modified Figure 2 as well as labeled the axis. We hope that the reviewer agree that this version is now suitable for publication.

Round 3

Reviewer 1 Report

Authors have modified as suggested. 

Hence this reviewer indicate accept this manuscript for. Publication in Pharmaceuticals